# Hybrid Prediction Model Based on Decomposed and Synthesized COVID-19 Cumulative Confirmed Data

**Zongyou Xia [1], Gonghao Duan [2,\*] and Ting Xu [3]**

1   School of Computer Science and Engineering, Wuhan Institute of Technology, Wuhan 430205, China; 22107010124@stu.wit.edu.cn
2   Hubei Key Laboratory of Advanced Control and Intelligent Automation for Complex Systems, Wuhan 430074, China
3   Engineering Research Center of Intelligent Technology for Geo-Exploration, Ministry of Education, Wuhan 443002, China; 2006310123@stu.wit.edu.cn
\*   Correspondence: duangh@wit.edu.cn

**Abstract:** Since 2020, COVID-19 has repeatedly arisen around the world, which has had a significant impact on the global economy and culture. The prediction of the COVID-19 epidemic will help to deal with the current epidemic and similar risks that may arise in the future. So, this paper proposes a hybrid prediction model based on particle swarm optimization variational mode decomposition (PSO-VMD), Long Short-Term Memory Network (LSTM) and AdaBoost algorithm. To address the issue of determining the optimal number of modes $K$ and the penalty factor ($\alpha$) in the variational mode decomposition (VMD), an adaptive value for particle swarm optimization (PSO) is proposed. Specifically, the weighted average sample entropy of the relevant coefficients is utilized to determine the adaptive value. First, the epidemic data are decomposed into multiple modal components, known as intrinsic mode functions (IMFs), using PSO-VMD. These components, along with policy-based factors, are integrated to form a multivariate forecast dataset. Next, each IMF is predicted using AdaBoost-LSTM. Finally, the prediction results of all the IMF components are reconstructed to obtain the final prediction result. Our proposed method is validated by the cumulative confirmed data of Hubei and Hebei provinces. Specifically, in the case of cumulative confirmation data, the coefficient of determination ($R^2$) of the mixed model is increased compared to the control model, and the average mean absolute error (MAE) and root mean square error (RMSE) decreased. The experimental results demonstrate that the VMD–AdaBoost–LSTM model achieves the highest prediction accuracy, thereby offering a new approach to COVID-19 epidemic prediction.

**Keywords:** COVID-19 epidemic; prediction; variational mode decomposition

## 1. Introduction

In late 2019, an outbreak of novel coronavirus pneumonia occurred in Wuhan, Hubei Province, China. The virus spread rapidly around the world in a short period of time. It has posed enormous threats and challenges to human life, global public health security, and the global economy, causing significant impacts and damage. Targeted epidemic prevention policies and measures are key to controlling the spread of the epidemic. Accurate predictions of the spread of the epidemic can provide a reference for countries to design epidemic prevention policies. Therefore, how to improve prediction accuracy has become the focus of research for many scholars [1–3].

### 1.1. Literature Review

Since the outbreak of the epidemic, many scholars have carried out research on the prediction of COVID-19, and this can be roughly divided into two categories: (1) single model prediction, and (2) hybrid model prediction. Single-model forecasting can be divided into traditional epidemiological models and machine learning models. In [3], the SIR model

was used to predict the COVID-19 epidemic, but the results were not satisfactory, because the development of the epidemic was affected by many factors, and it was difficult to achieve accurate prediction results using the pure Susceptible Infected Recovered (SIR) model [4]. In [5], an SIR model that can adapt to the surge of susceptible populations is developed. It can better predict the actual number of infections and deaths in some areas, but it still struggles to overcome the sudden change in external factors. It is challenging to accurately predict the drastic fluctuations in the actual numbers of infections and deaths due to changing external factors. In [6–9], the traditional Susceptible, Exposed, Infected, Recovered (SEIR) model was combined with some external factors such as quarantine, vaccination and other strategies for the prediction of the COVID-19 epidemic, and they achieved good results, but the parameters of the model, such as the mutation rate of the virus, the probability of recovery, etc., change dynamically during the evolution of the epidemic, and these models do not take this feature into account. So, there are still some problems.

In [10], four models were used to predict the COVID-19 epidemic—cumulative confirmed cases and deaths, including linear regression, exponential smoothing, least absolute shrinkage and selection operator (Lasso), and support vector machine (SVM). Among them, the predictive effect of SVM applied to the three types of data is poor. Ref. [11] used random forest, linear regression, and multi-layer perceptron to predict data for the United States, Germany, and the world. Among these models, SVM achieved the best results. This is due to the fact that the epidemic data are inherently unstable and difficult to predict. If the complexity of the data can be reduced, the prediction effect of SVM will be improved.

In [12], the LSTM model was first used to predict the spread of the epidemic in Canada, Italy, and the United States. In [13], LSTM, bidirectional long short-term memory (BILSTM), and Encoding–Decoding–LSTM models were used to predict the development of the epidemic in India. Among these models, Encoding–Decoding–LSTM achieved the best prediction effect. In [14], BILSTM–Encoding–Decoding and 16 other models were used to make comparative predictions based on the epidemic data from Saudi Arabia. In the end, BILSTM–Encoding–Decoding achieved the best results, but a problem was also raised in [14], in that it failed to consider the impact of relevant policies on the development of the epidemic.

In [15], the Autoregressive Integrated Moving Average (ARIMA) model was used to compare SIR, SEIR and other infectious disease models, and to predict the epidemic situation in Kuwait. Finally, the ARIMA model achieved better results. In [16], ARIMA was used to predict the epidemic trend pattern in Saudi Arabia. The best ARIMA model for predicting daily new cases was identified, enabling the estimation of the approximate number of confirmed cases in the next month.

To sum up, the infectious disease model can achieve better prediction results when there is a large amount of stable data, but the epidemic data are complex, nonlinear, and unstable. Moreover, the infectious disease model can only make rough predictions, and cannot achieve accurate predictions. Single models such as SVM, ARIMA, and LSTM can accurately predict the development of the epidemic, but they also have certain limitations.

To achieve better prediction accuracy, some researchers have proposed a hybrid prediction model, combining different models to achieve the desired forecasting effect. Hybrid models can be roughly divided into two categories: hybrid models based on component composition, and hybrid models based on data processing. The former combines more than two models to predict the epidemic. For example, in [17], a combination of the hybrid neural network and SEIR model predicted that the occupancy rate of ICUs during the epidemic would improve at the national level and within the city. While the superposition of two models may improve the prediction accuracy to a certain extent, this may not necessarily overcome the limitations of the model itself. Furthermore, introducing too many models may increase the number of undetermined parameters and complicate the forecasting process. The latter approach is built upon a hybrid model of data processing, using the related processing of input data, such as decomposition, filtering, denoising,

feature engineering, interpolation, and other processing methods. The decomposition method can decompose the original data into components of varying complexity, which may more effectively capture the inherent characteristics of the data, reduce the complexity of the original sequence, and ultimately yield more accurate prediction results. For example, ref. [18] proposed a prediction method based on wavelet decomposition (DWD). He used DWD to decompose the original sequence into several sub-components, each of which contained some of the information of the original sequence, and he then used ARIMA to predict the sub-components. Finally, the prediction results of the subcomponents were linearly summed as the final prediction result. Good results were obtained in most cases, but the nature of DWD is a Fourier transform, which is difficult to apply to non-stationary signals such as epidemics.

Ref. [19] proposed an EEMD-based prediction model for the COVID-19 epidemic, which decomposes the input raw data into several modal components, and then uses LSTM to make predictions on these components. The data-based processing model is more adaptable to various types of data, leading to prediction results that are closer to the actual data. However, empirical mode decomposition algorithms such as Empirical Mode Decomposition (EMD), Ensemble Empirical Mode Decomposition (EEMD), and Complete Ensemble Empirical Mode Decomposition with Adaptive Noise (CEEMDAN) have common shortcomings, such as a lack of a solid theoretical foundation and modal aliasing, meaning that similar frequencies are decomposed into separate components, which can lead to significant deviations from forecasts. VMD is a cutting-edge decomposition algorithm with a solid theoretical foundation, capable of addressing the issue of modal aliasing [20] and effectively capturing inherent data characteristics. Therefore, to avoid the limitations of other decomposition algorithms, this paper chooses VMD to decompose the original data [21]. However, VMD also has its own disadvantages. The modal component $K$ and the penalty factor $\alpha$ are determined based on experience and are relatively subjective [22]. To address this problem, the PSO algorithm is introduced. PSO algorithm is a classic intelligent optimization algorithm. It is easy to implement and achieves relatively fast convergence. Therefore, this paper proposes to use PSO algorithm to obtain $K$ and a penalty factor, so as to improve the adaptability of the algorithm.

In addition, Refs. [23,24] described positive intervention methods, such as wearing masks and home isolation, which have a significant curbing effect on the development of the epidemic. However, due to the difficulty in obtaining data, we only introduced policy factors to simulate the actual epidemic situation. The introduction to policy factors will inevitably increase the difficulty of forecasting. To improve the accuracy of forecasting, we propose a hybrid prediction model, PSO–VMD–AdaBoost–LSTM, based on optimized variational mode decomposition (PSO-VMD) and an AdaBoost-enhanced long short-term memory network (AdaBoost-LSTM).

*1.2. Contributions of This Paper*

1.  Currently, COVID-19 predictions are mostly univariate. This paper builds a new predictive dataset by incorporating policy factors.
2.  Existing hybrid models for COVID-19 prediction are mainly based on component combinations, while methods based on data processing have just begun to emerge. Aiming at the defects of EMD and its optimization algorithms such as EEMD, CEEMDAN, etc., this paper proposes a VMD method for decomposing COVID data.
3.  The VMD method has certain defects, namely, the key parameters $K$ and $\alpha$ need to be preset based on experience. Aiming at this problem, this paper proposes a PSO–VMD method, which realizes the adaptive value of $K$ and $\alpha$.
4.  The AdaBoost algorithm is proposed to enhance the LSTM, which finally improves the prediction accuracy to a certain extent.
5.  The effectiveness of the method is verified and tested using the epidemic data from Hubei Province and Hebei Province.

## 2. Materials and Methods

### 2.1. Variational Mode Decomposition (VMD)

VMD is a novel adaptive non-recursive signal decomposition method that enhances the stability of sequences [25]. The VMD method can effectively overcome the mode mixing issue of the EMD method [26] and has a solid theoretical foundation. In this study, VMD is used to decompose the COVID-19 data, and the structure and solution of VMD are as follows.

The structure of VMD involves constructing and solving a variational constraint problem to decompose a sequence. The epidemic time series is treated as the input signal *f*, and the variational problem is described as using the center frequency to find *K* modal functions with limited bandwidth to minimize the sum of the bandwidth estimates of each mode [26]. The model of the constructed variational constraint problem is formulated as in Formula (1).

$$
\min_{\{u_k\},\{\omega_k\}} \left\{ \sum_{k=1}^{K} \left\| \partial_t \left[ \left( \delta(t) + \frac{j}{\pi t} \right) u_k(t) \right] \ell^{-j\omega k \mathrm{t}} \right\|_2^2 \right\}
$$
$$
s.t. \sum_{k=1}^{K} u_k(t) = f(t)
$$
(1)

Formula (1) shows the intrinsic mode function (IMF) obtained after decomposition. $\{\omega_k\} = \{\omega_1,\dots,\omega_k\}$ represents the corresponding central frequency of each IMF. $\delta(t)$ is the impact function. To solve the variational constraint problem described above, the Lagrange multiplication operator $\lambda(t)$ is introduced to penalize the factor *C*. This converts the constrained variational problem into an unconstrained one, expressed as Formula (2), which enables the solving of the variational problem.

$$
L(\{u_k\},\{\omega_k\},\lambda) = C \sum_{k=1}^{K} \left\| \partial_t \left[ \left( \delta(t) + \frac{j}{\pi t} \right) u_k(t) \right] \ell^{-j\omega k \mathrm{t}} \right\|_2^2
$$
$$
+ \left\| f(t) - \sum_{k=1}^{K} u_k(t) \right\|_2^2 + \left\langle \lambda(t), f(t) - \sum_{k=1}^{K} u_k(t) \right\rangle
$$
(2)

The Alternate Direction Method of Multipliers (ADMM) is introduced, where $u_k^{n+1}, \omega_k^{n+1}, \lambda_k^{n+1}$ are alternately updated, and n represents the number of iterations.

$$
\hat{u}_k^{(n+1)}(\omega) = \frac{\hat{f}(\omega) - \sum_{k=1}^{K} \hat{u}_k(\omega) + \frac{\lambda(\omega)}{2}}{1 + 2C(\omega - \omega_k)}
$$
(3)

$$
\hat{\omega}_k^{(n+1)} = \frac{\int_0^\infty \omega |\hat{u}_k(\omega)|^2 \mathrm{d}\omega}{\int_0^\infty |\hat{u}_k(\omega)|^2 \mathrm{d}\omega}
$$
(4)

$$
\hat{\lambda}^{n+1}(\omega) = \hat{\lambda}^{n+1}(\omega) + \tau \left[ \hat{f}(t) - \sum_{k=1}^{K} u_k^{n+1}(\omega) \right]
$$
(5)

The iteration is terminated based on the accuracy *e*, and the stopping condition is expressed in Formula (6).

$$
\sum_{k=1}^{K} \frac{\left\| \hat{u}_k^{(n+1)} - \hat{u}_k^n \right\|_2^2}{\left\| \hat{u}_k^n \right\|_2^2} < e
$$
(6)

### 2.2. LSTM

LSTM is carefully designed by adding memory features based on Recurrent Neural Networks (RNN) to avoid the problems of long-term dependence [27]. LSTM can maintain the long-term memory of the neural network, making the model more suitable for epidemic data prediction. For RNN, due to the unrestricted updating of information in the network layer, there will be a problem of gradient disappearance or gradient explosion, while the

LSTM network introduces forgetting units and storage units into the hidden layer, which discards secondary information when new information is input and retains important information in long-term memory. These units are referred to as gates in the LSTM model.

Forget gate: read the output $h_{t-1}$ of the previous layer and the current input $x_t$, output and assign it to the current cell state $C_{t-1}$.

$$f_t = \sigma(W_f[h_{t-1}, x_t] + b_f) \tag{7}$$

Input gate: remember the current $i_t$ and part of the output information $\widetilde{C}_t$ of the forget gate.

$$i_t = \sigma(W_i[h_{t-1}, x_t] + b_i) \tag{8}$$

$$\widetilde{C}_t = \tanh(W_c[h_{t-1}, x_t] + b_C) \tag{9}$$

Output gate: select a part of the cell state and finally output $h_t$.

$$\sigma_t = \sigma(W_o[h_{t-1}, x_t] + b_o) \tag{10}$$

$$h_t = \sigma\tanh(C_t) \tag{11}$$

The $\sigma$ is the sigmoid activation function.

In Figure 1, each solid black line transmits the entire vector, the circles represent element-wise operations, and the rectangular boxes represent network layers. The composite lines represent connections between vectors.

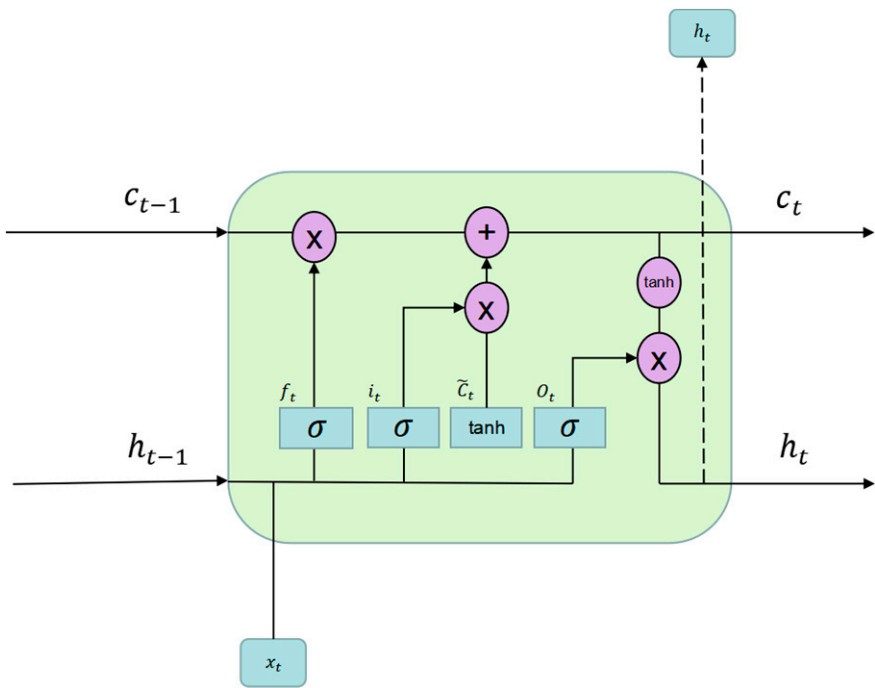

**Figure 1.** LSTM memory cell.

### 2.3. VMD Optimized by PSO

If the value of $K$ is too large or too small, it will affect the final prediction. So, this paper proposes a VMD method based on particle swarm optimization (PSO). Inspired by [28], a weighted average sample entropy based on the correlation coefficients is proposed as the objective function for the PSO algorithm to optimize VMD and achieve better decomposition results. The correlation coefficient is a measure of the degree of association between two signals [29].

Calculation method of correlation coefficient: $x_i$ is a subsequence, $y_i$ is the original sequence, $\overline{x}_i$ is the mean of the subsequence, $\overline{y}_i$ is the mean of the original series, and $K$ is the number of the IMF.

$$r_i = \frac{\sum\limits_{i=1}^{K} (x_i - \overline{x_i})(y_i - \overline{y})}{\sum\limits_{i=1}^{K} (x_i - \overline{x_i})^2 (y_i - \overline{y})^2} i = 1, 2 \ldots K \tag{12}$$

By calculating the $r_i$ between each IMF and the original signal, we can assign weights to the sample entropy of each IMF. The overall average sample entropy is then calculated for each $K$ value, which serves as the fitness value for the particle swarm optimization algorithm. Therefore, the fitness function of the particle swarm algorithm is $f_{obj}$. Here, SampEn is used to calculate the sample entropy of the current, and $r_i$ to calculate the correlation coefficient between the current and the original sequence. Formula (13) is the objective function of PSO. Figure 2 introduces the process of PSO for optimizing VMD.

$$f_{obj} = \frac{\sum\limits_{i=1}^{K} \text{SampEn}(IMF_i) r_i}{K}, i = 1, \ldots, K \tag{13}$$

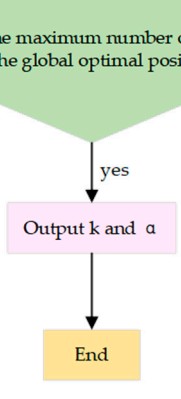

**Figure 2.** The process of PSO–VMD.

## 2.4. AdaBoost–LSTM

Due to the inherent complexity and randomness of epidemics, a simple LSTM model is insufficient for accurate prediction. Thus, the AdaBoost algorithm is used to enhance the LSTM network.

The fundamental idea of the AdaBoost algorithm is to aggregate the training results of multiple weak learners to obtain a strong learner capable of tackling complex tasks, with the strong learner's prediction results serving as the final output [30].

While originally developed for classification tasks, AdaBoost could also be adapted for regression problems. The specific method of using the AdaBoost algorithm to enhance the LSTM neural network model is given below. Figure 3 is the framework of Adaboost-LSTM.

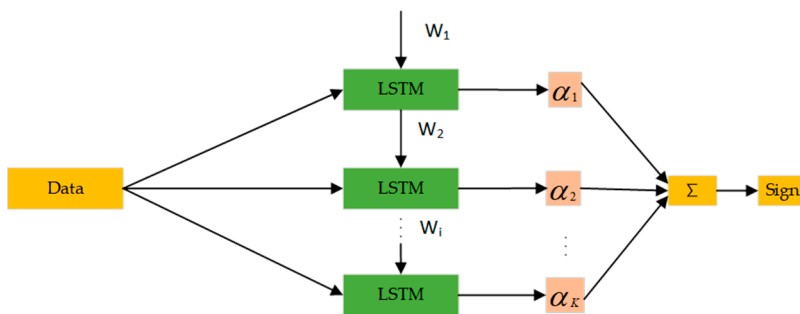

**Figure 3.** AdaBoost framework.

The specific algorithm flow is as follows:

Step 1. Initialize the distribution of weights for the training data. Each sample is given the phase $W_I = \frac{1}{N}$, and $N$ is the total number of samples. For each sample after the first iteration, the weight set of $D_i = (W_{11}, W_{12}, \ldots, W_{1N})$;

Step 2. Set the number of weak learners $K$. According to time and accuracy requirements, comprehensively set the number of weak learners;

Step 3. Weak learner prediction—train the $i$-th weak predictor, use the training data to predict in the LSTM network, get the output result, and compare it with the real value to get the error function $e_i$ . The calculation method is

$$e_t = \sum_i D_i, i = 1, 2, \ldots, N \tag{14}$$

Step 4. Compute weak predictor weights. Calculate weak according to error function value. The predictor weight is $\alpha_t$ , and the calculation formula is

$$\alpha_t = \frac{1}{2} \ln \left( \frac{1 - e_i}{e_i} \right), t = 1, 2, \ldots, N \tag{15}$$

Step 5. Update the training data sample weights. Adjust the training data sample weight by the weak predictor weight *a*. The update formula is

$$W_{t+1,i} = \frac{\omega_{t,i}}{\beta} \exp(-a_t y_i f_t(x_i)) \tag{16}$$

Step 6. Strong predictive function. After the weak learner is trained $K$ times, $K$ groups of weak prediction values are obtained, and the strong predictor is obtained after combination. The calculation formula of the strong predictor is

$$G(x) = sign \left[ \sum_{t=1}^{K} a_g f_t(x) \right] \tag{17}$$

*2.5. Final Model*

Based on the theoretical foundations above, we propose a hybrid prediction model, named PSO–VMD–AdaBoost–LSTM, that combines PSO, VMD, LSTM, and AdaBoost. The workflow of the proposed model is depicted in Figure 4.

First, the COVID-19 data are optimized using PSO–VMD to iteratively obtain the optimal $K$ value and alpha, as well as the decomposed intrinsic mode functions. Based on the obtained IMF components, the AdaBoost–LSTM prediction model is established to generate the output results $pre_i$. Finally, the prediction results of each component are

combined as in Formula (18), as the complete prediction result, and *K* is the number of the IMF.

$$X_{mix} = \sum_{i=1}^{n} pre_i, i = 1, 2 \ldots K \tag{18}$$

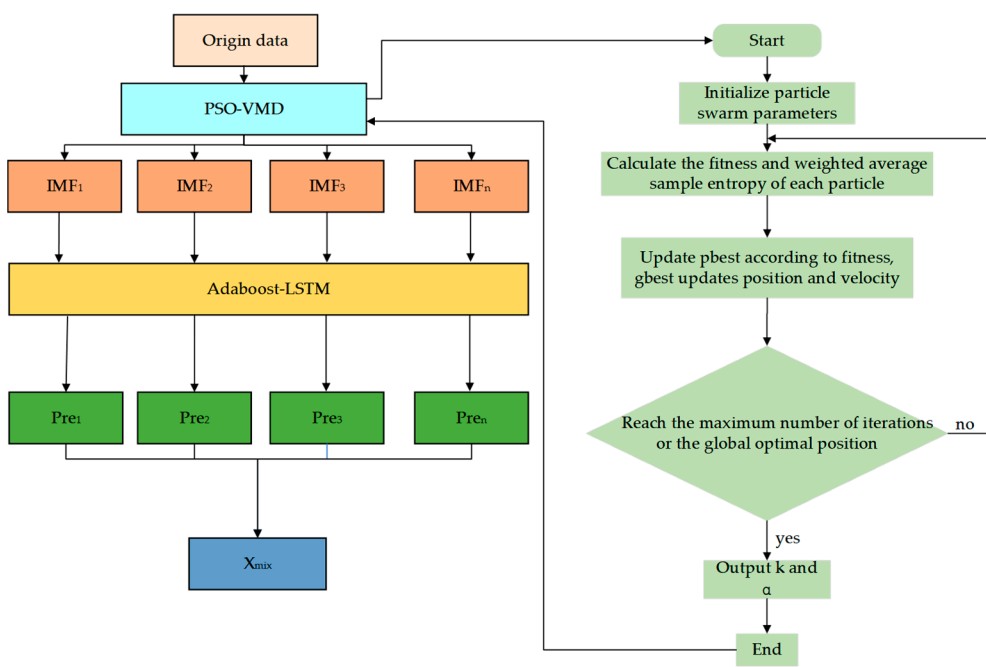

**Figure 4.** The flow chart of PSO–VMD–AdaBoost–LSTM.

*2.6. Date Modeling*

2.6.1. Data Sources and Experimental Environment

This study has conducted a predictive simulation experiment using actual daily cumulative confirmed COVID-19 data. The data of Hubei Province and Hebei Province from 1 May 2020 to 31 December 2021 were selected from the COVID-19 policy tracking data warehouse released by Oxford University "https://github.com/OxCGRT/covid-policy-tracker (accessed on 13 April 2020)", which is a multivariate time series with multiple dependent variables over time. In the end, a category of variables with strong correlation was selected, comprising mainly those related to policies that curbed public activities and closed public places and transportation. The variables are explained in Table 1. Finally, the first 90% of the data were selected as the training set, and the last 10% as the testing set. The specific experimental environment was developed using Python 3.7. The CEEMDAN decomposition was performed using the PyEMD library version 1.2.3, while the VMD decomposition used the vmdpy library. The LSTM network was built by Tensorflow 2.6. The cumulative confirmed data of Hubei Province and Hebei Province are shown in Figure 5.

**Table 1.** Policy variable parameter description.

| Variable | Explain |
|---|---|
| C1_School closing | Record school suspensions |
| C2_Workplace closing | Record the closure of public places |
| C3_Cancel public events | Record the development of public events |
| C4_Restrictions on gatherings | Record party restrictions |
| C5_Close public transport | Record whether public transport is closed |
| C6_Stay at home requirements | Record whether home isolation is imposed |
| C7_Restrictions on internal movement | Record restrictions on internal mobility between cities/regions |

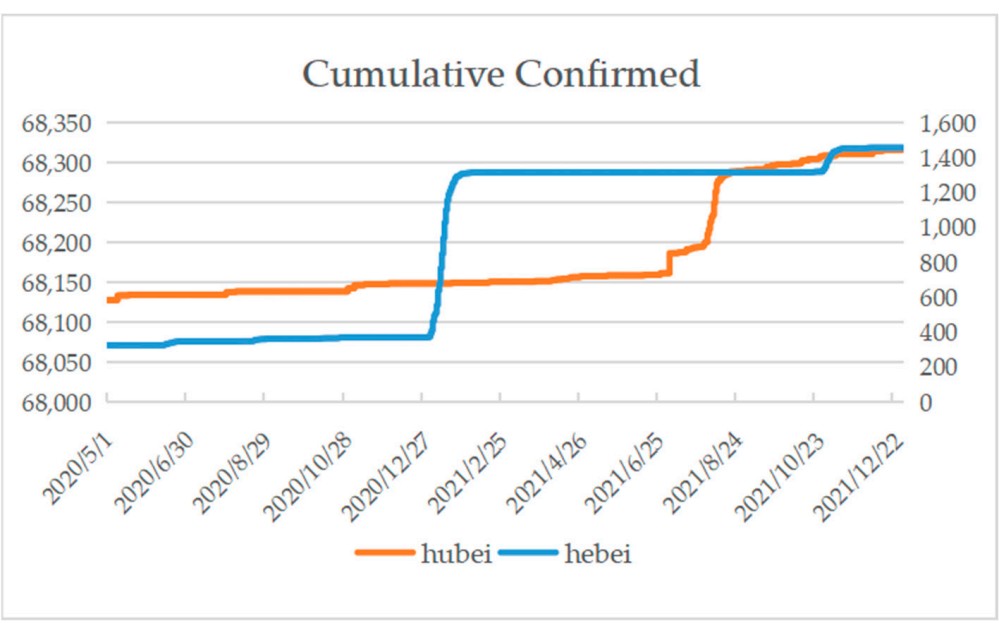

**Figure 5.** Cumulative confirmed cases.

2.6.2. Evaluating Indicator

To quantitatively describe and evaluate the performance of the model, root mean square error (*RMSE*), the coefficient of determination ($R^2$), the mean absolute error (*MAE*), and the mean absolute percentage Error (*MAPE*) are selected.

$$RMSE = \sqrt{\frac{1}{n}\sum_{i=1}^{n}(Y_i - \hat{Y}_i)^2} \tag{19}$$

$$R^2 = 1 - \frac{\sum_{i=1}^{n}(\hat{Y}_i - Y_i)^2}{\sum_{i=1}^{n}(\overline{Y}_i - Y_i)^2} \tag{20}$$

$$MAE = \frac{1}{n}\sum_{i=1}^{n}|Y_i - \hat{Y}_i| \tag{21}$$

$$MAPE = \frac{1}{n}\sum_{i=1}^{n}\frac{|Y_i - \hat{Y}_i|}{Y_i} \tag{22}$$

$Y_i$ is the true value, $\hat{Y}_i$ is the predicted value, and *n* is the total number of samples.

## 3. Results

### 3.1. The Experiment of PSO–VMD

Since the particle swarm optimization algorithm was proposed, many scholars have carried out relevant research. To improve the convergence performance of the algorithm, we introduced the concept of inertia weight *w*. The inertia weight (*w*), as a crucial parameter in PSO, determines the magnitude of influence exerted by the historical state on the algorithm's evolutionary process. This paper introduces a strategy of non-linear weight reduction, as shown in Formula (23).

$$\omega_O = \omega_{\text{start}} - (\omega_{\text{start}} - \omega_{\text{end}}) \times \left(2 \times (p/T_{\max}) - (p/T_{\max})^2\right) \tag{23}$$

$\omega_{\text{start}}$ is the initial weight, $\omega_{\text{end}}$ is the termination weight, *p* is the current round number of iteration, and $T_{\max}$ is the maximum number of iteration rounds. The dynamically

decreasing $\omega$ enables the algorithm to conduct a global search in the early stage of operation. This helps in finding more optimal solutions in a large search space. The gradually reducing $\omega$ in the iterative process gives the particles a finer local search ability, which is beneficial in helping the algorithm to search for a better optimal solution among the potential optimal solutions that have been identified. Therefore, we set the initial weight of the particle swarm optimization algorithm to 0.9, and gradually decreased the termination weight by 0.1; the step size of each decrease was also 0.1.

The cumulative confirmed data in Hubei Province from 1 May 2020 to 31 December 2021 have been used as the input of the VMD algorithm. The particle swarm algorithm with inertia weight was used to optimize VMD. Table 2 shows the other parameters of the particle swarm optimization algorithm. Figure 6 shows the decrement in inertia weight across eight experiments. The results of the final eight iterations are all 4, and $\alpha = 2685 \pm 10$. Therefore, the particle swarm optimization algorithm can be optimal. The final decomposition result of PSO–VMD is shown in Figure 7.

**Table 2.** Parameters of particle swarm algorithm.

| Variable | Value |
| --- | --- |
| population | 30 |
| $T_{max}$ | 50 |
| acceleration constant | 2 |
| boundary | $K \in [2, 12]$, $\alpha \in [10, 5000]$ |

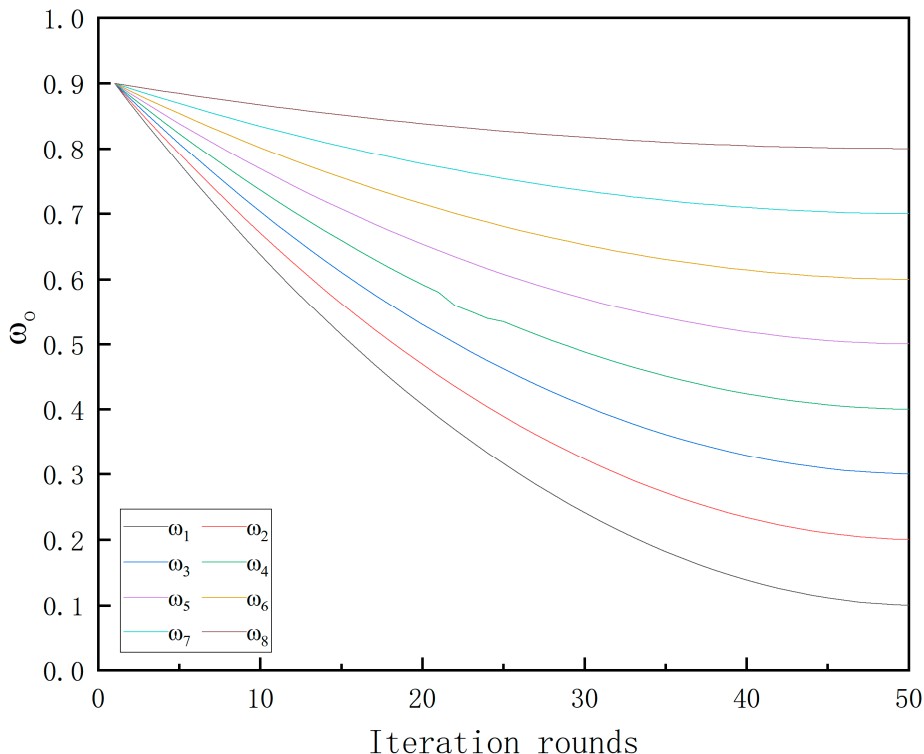

**Figure 6.** Decrement in inertia weight in the iterative process of PSO.

To compare the effects of VMD decomposition, the complete set of empirical mode decompositions was selected for comparison. CEEMDAN is an optimization algorithm based on empirical mode decomposition, which improves the defects of the EMD algorithm to a large extent. CEEMDAN is widely used in many fields due to its high performance ability. Figure 8 is the decomposition result of CEEMDAN.

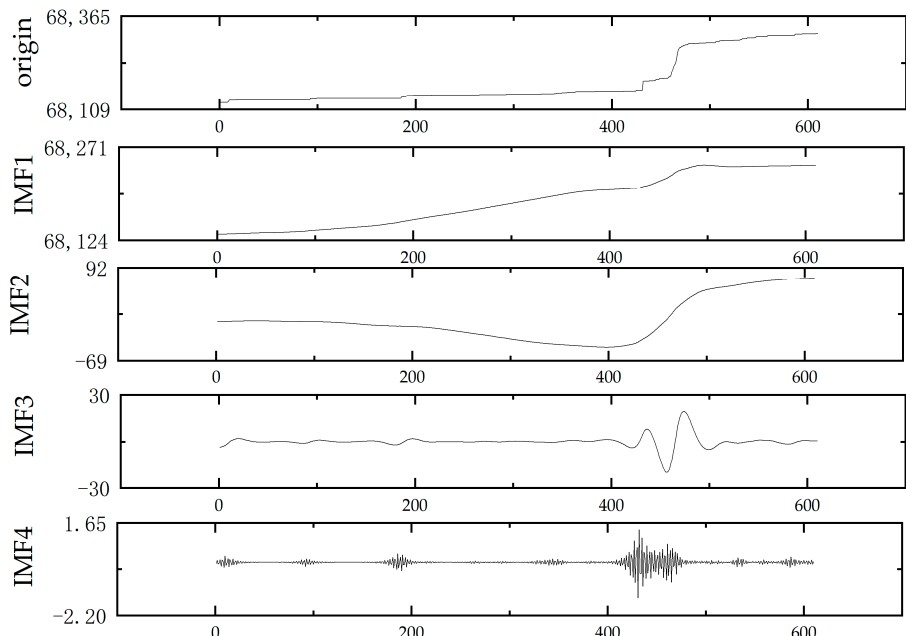

**Figure 7.** Results of VMD decomposition.

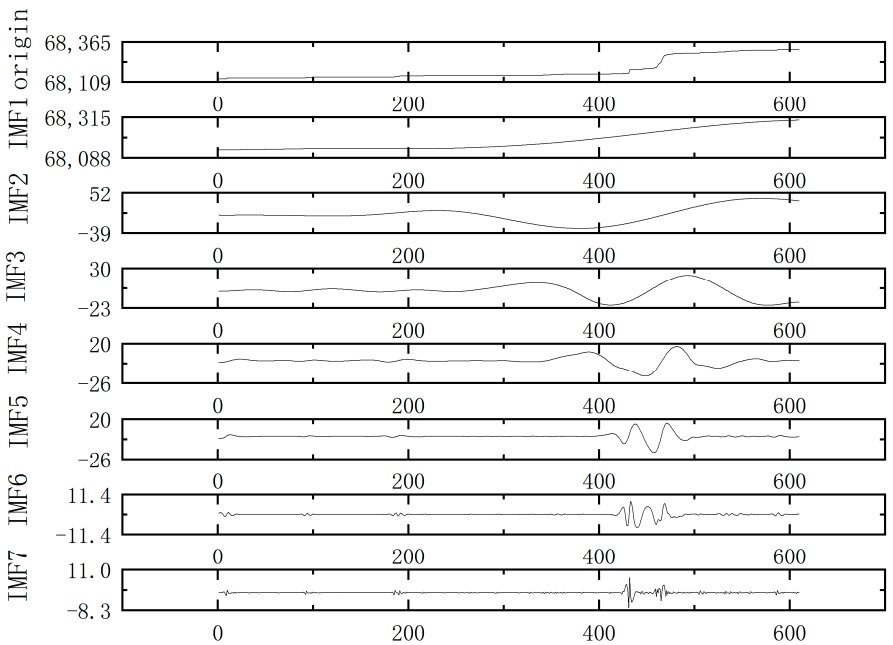

**Figure 8.** Results of CEEMDAN decomposition.

In Figure 8, IMF4 and IMF5 exhibit obvious modal aliasing phenomena. These two modes are of the same characteristic scale, and their waveforms are essentially similar, which satisfies the modal aliasing standard proposed by Huang [31]. In Figure 8, the first three components show a significant swing on the right end due to the noise interference, and these three components account for a large proportion of the original sequence, so they will have a greater impact on the entire sequence. The PSO–VMD method shown in Figure 7 demonstrates smaller waveform distortion and better noise robustness.

### 3.2. Forecasting Results

3.2.1. Prediction Based on Hubei Cumulative Confirmed Data

In this chapter, we present the prediction results obtained from the previously discussed VMD–AdaBoost–LSTM model. Table 3 is the four evaluation indicators mentioned

in Section 2.6.2, which are used for quantitative analysis.The key parameters of the model are shown in Table 4. To demonstrate its superiority, it has been compared with three other models: CEEMDAN–LSTM, VMD–LSTM and LSTM. Figure 9 shows the comparison of the prediction results of the four prediction models. For example, Figure 10 presents the comparison results of the overall model using the data from Hubei Province. Figure 11 is the error distribution diagram of the four models.

**Table 3.** Evaluation index of each model.

| Model | $R^2$ | *RMSE* | *MAE* | *MAPE* |
|---|---|---|---|---|
| LSTM | 0.89 | 0.8740 | 0.43069 | 0.00063 |
| CEEMDAN–LSTM | 0.79 | 1.2474 | 1.12655 | 0.00165 |
| VMD–LSTM | 0.93 | 0.6911 | 0.48259 | 0.00071 |
| VMD–AdaBoost–LSTM | 0.95 | 0.5979 | 0.40190 | 0.00059 |

**Table 4.** Model parameter settings.

| Method | Parameters | Values |
|---|---|---|
| LSTM | Layer | 2 |
| | Neurons | 64 |
| | Learning rate | 0.01 |
| | Batchsize | 32 |
| | Epoch | 50 |
| VMD–LSTM | Layer | 3 |
| | Neurons | 128 |
| | Learning rate | 0.001 |
| | Batchsize | 32 |
| | Epoch | 60 |
| CEEMDAN–LSTM | Layer | 2 |
| | Neurons | 256 |
| | Learning rate | 0.002 |
| | Batchsize | 64 |
| | Epoch | 50 |
| VMD–AdaBoost–LSTM | Layer | 3 |
| | Neurons | 128 |
| | Learning rate | 0.001 |
| | Batchsize | 32 |
| | Epoch | 100 |
| | Number of base learners | 5 |

It can be seen from Figure 9 that, compared with the other three models, the prediction effect of VMD–AdaBoost–LSTM is most consistent with the original data. Furthermore, it can be seen from Figure 10 that the predicted points of VMD–LSTM and VMD–AdaBoost–LSTM are the closest to the original point. In Figure 11, it is shown that the predicted value of VMD–AdaBoost–LSTM fluctuated slightly around the actual value in the first 32 days, and the error increased in the middle 10 days. However, in the following 20 days, it gradually stabilized. Finally, combined with the evaluation index shown in Table 3, the VMD–AdaBoost–LSTM proposed in this paper has the best prediction effect, with a maximum error of 2.06 and a minimum error of −1.23. VMD–AdaBoost–LSTM predicts the least overall fluctuation compared with the other three models. The predictive performance of VMD–AdaBoost–LSTM is clearly superior to that of other models.

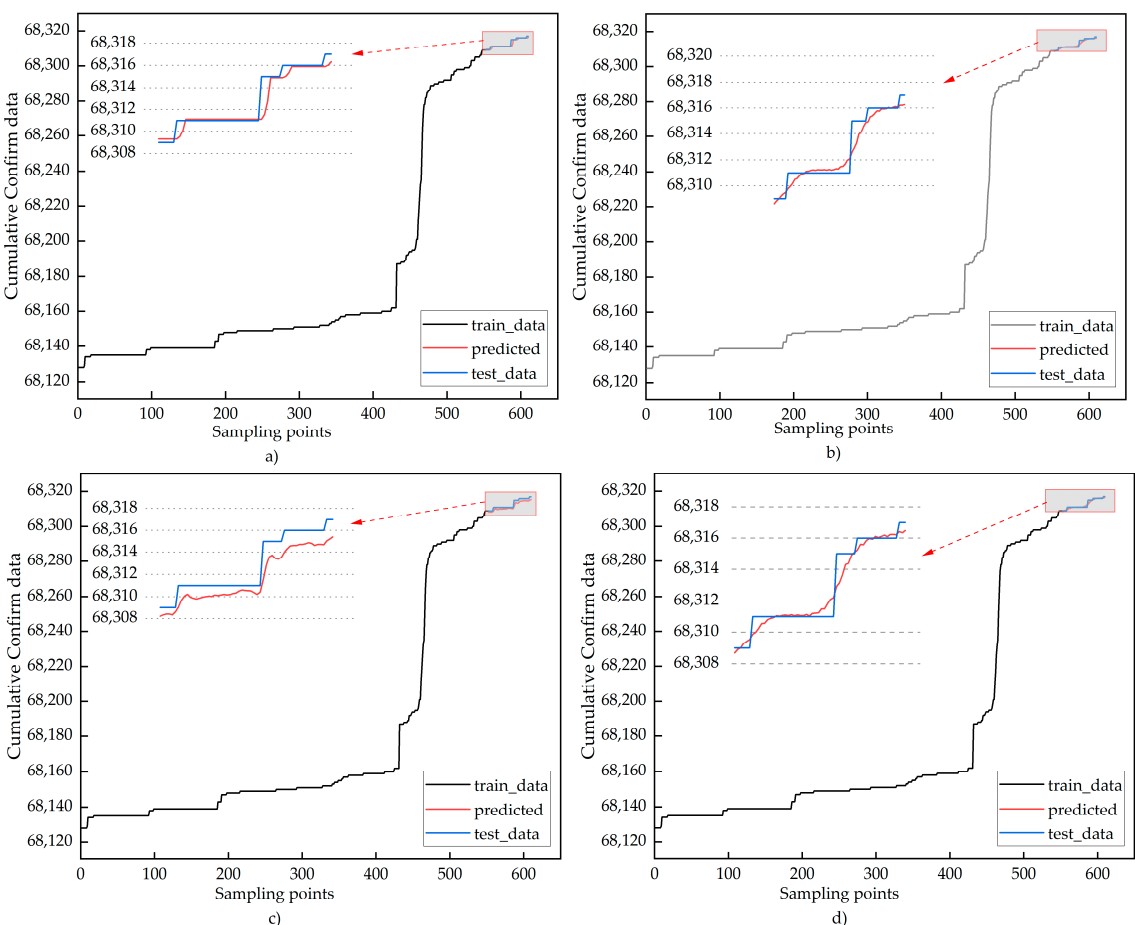

**Figure 9.** (**a**) The prediction result of LSTM in Hubei. (**b**) The prediction result of VMD–LSTM in Hubei. (**c**) The prediction result of CEEMDAN–LSTM in Hubei. (**d**) The prediction result of VMD–AdaBoost–LSTM in Hubei.

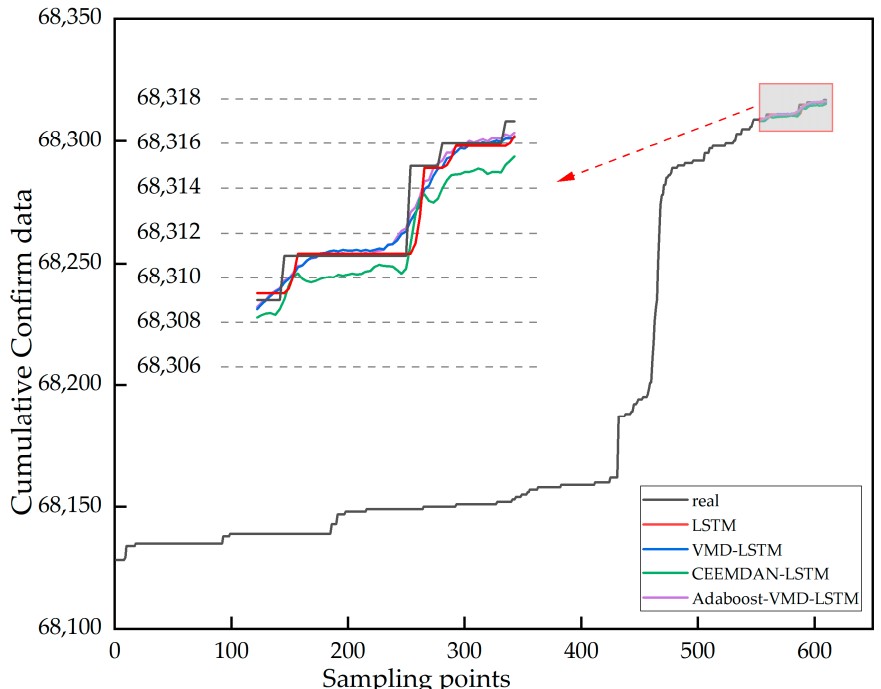

**Figure 10.** Prediction results of each model.

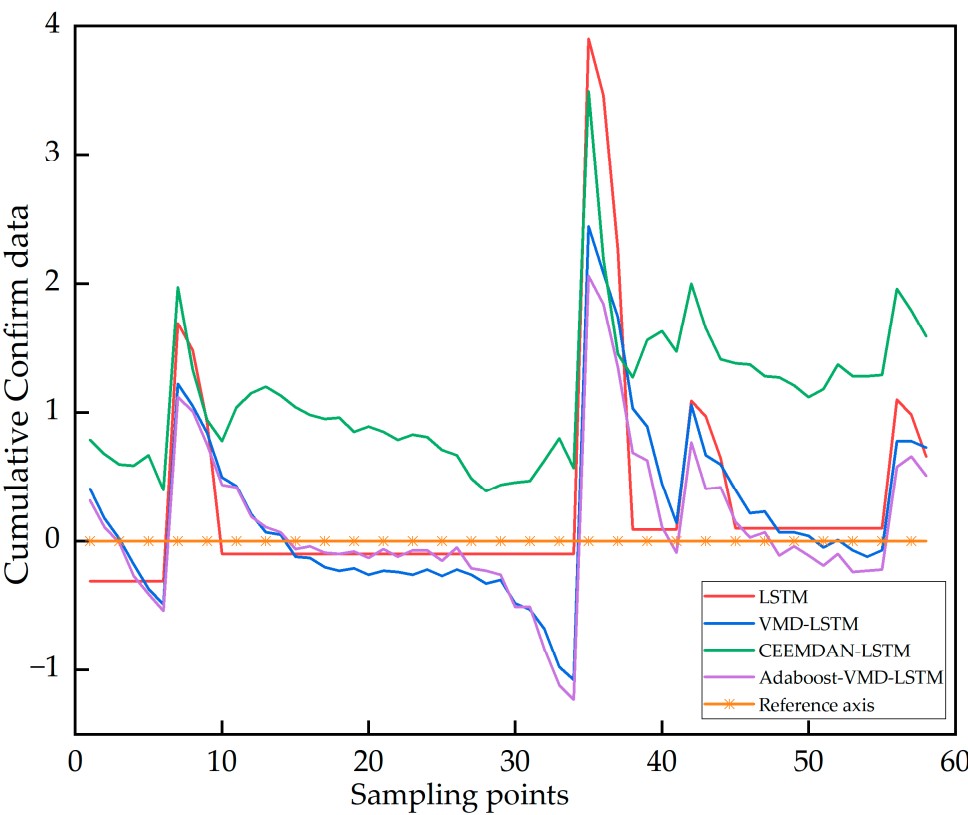

**Figure 11.** Distribution of Forecast Errors in Hubei.

The indicators of Table 3 show the average RMSE, MAE, and MAPE of VMD–AdaBoost–LSTM and the other three models. It was observed that the average RMSE of VMD–AdaBoost–LSTM decreased by 36.2%, while the average MAPE decreased by 80.2%. Additionally, the average MAE showed a reduction of 40%. Moreover, the R-square value of VMD–AdaBoost–LSTM was found to be the highest among all models.

Therefore, according to Figures 9–11, Table 3 offers the following conclusions:

1. The hybrid prediction model optimized by PSO–VMD demonstrates superior performance compared to the undecomposed model. This highlights the advantages of employing VMD decomposition in prediction.
2. The RMSE, MAE and MAPE of the hybrid model VMD–AdaBoost–LSTM are lower than those of the other three models. The prediction result of VMD–AdaBoost–LSTM is the best among all models. Hence, the utilization of the VMD method contributes to improved prediction accuracy.

### 3.2.2. Prediction Based on Hebei Cumulative Confirmed Data

This chapter will briefly introduce the experimental process of cumulative confirmed cases in Hebei Province and compare the prediction results of different models. The evaluation indicators of each model are presented in Table 5. Additionally, Figures 12 and 13 show the prediction effects of each model, among which VMD–AdaBoost–LSTM, LSTM, and VMD–LSTM exhibit relatively close prediction effects, while CEEMDAN has a relatively large prediction deviation. The results in Figure 14 indicate that the VMD–AdaBoost–LSTM model has a brilliant prediction effect, with the smallest deviation between the prediction curve and the real value. The evaluation indicators in Table 5 demonstrate that the evaluation indicators of VMD–LSTM and VMD–AdaBoost–LSTM are relatively close, and better than those of the other two models, indicating that the VMD decomposition method can effectively improve the accuracy of predictions. Meanwhile, the RMSE and MAE of VMD–LSTM and VMD–AdaBoost–LSTM are relatively close, indicating that the AdaBoost method cannot improve the performance of basic learners without limit. When

the performance of basic learners has reached a certain level or has been saturated, the impact of AdaBoost–LSTM on improvement will not be significant. To sum up, the VMD method can improve the prediction accuracy, and AdaBoost can also improve the prediction performance of base learning, which is consistent with the conclusion in Section 3.2.1.

**Table 5.** Evaluation index of each model.

| Model | $R^2$ | RMSE | MAE | MAPE |
|---|---|---|---|---|
| LSTM | 0.83 | 5.7447 | 4.0963 | 0.2844 |
| CEEMDAN–LSTM | 0.61 | 8.85517 | 6.21707 | 0.43007 |
| VMD–LSTM | 0.84 | 5.46402 | 4.08121 | 0.28175 |
| VMD–AdaBoost–LSTM | 0.91 | 4.2436 | 3.52603 | 0.24323 |

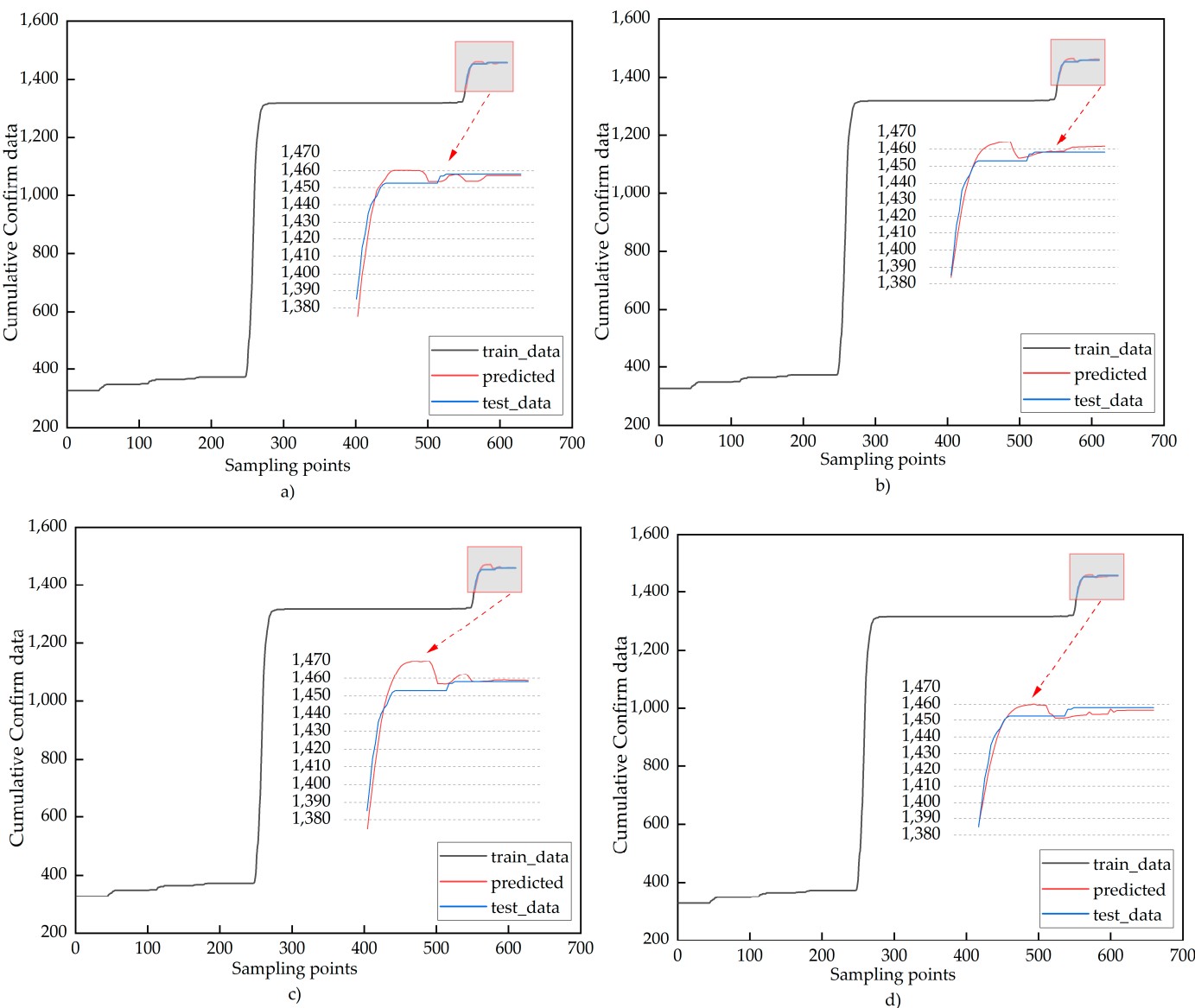

**Figure 12.** (**a**) The prediction result of LSTM in Hebei. (**b**) The prediction result of VMD–LSTM in Hebei. (**c**) The prediction result of CEEMDAN–LSTM in Hebei. (**d**) The prediction result of VMD–AdaBoost–LSTM in Hebei.

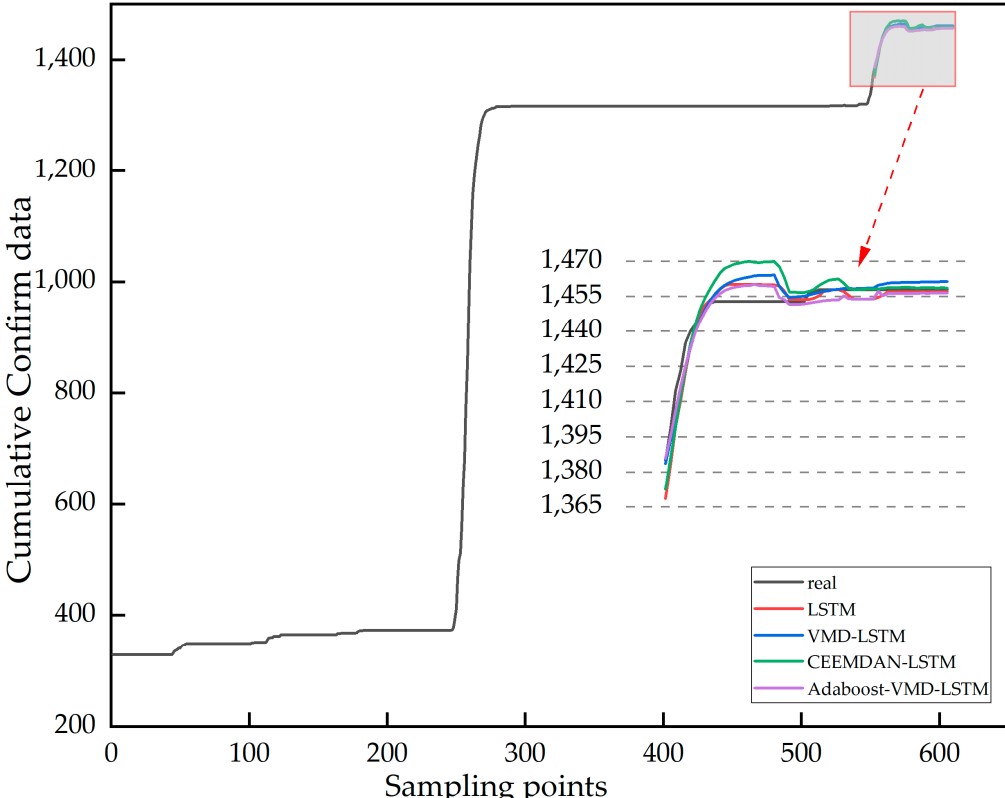

**Figure 13.** Comparison of prediction results of each model.

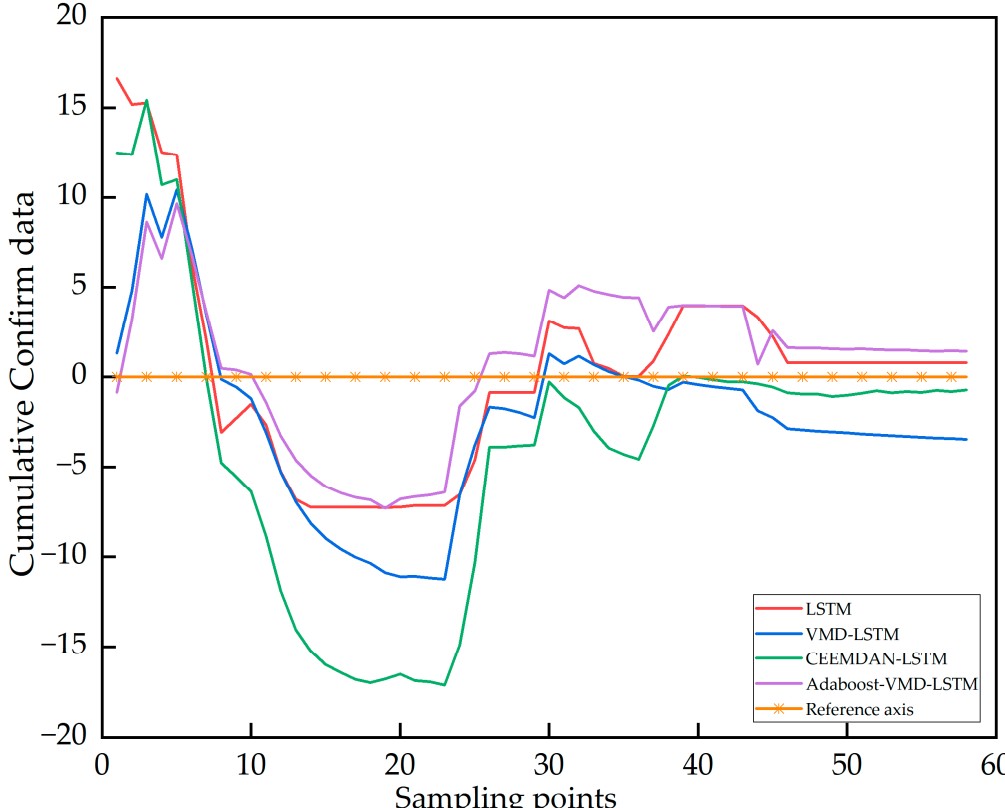

**Figure 14.** Distribution of Forecast Errors in Hebei.

## 4. Discussion

Although the epidemic situation has stabilized now, the research on the epidemic remains meaningful, as it provides insights for similar infectious diseases that may emerge in the future. The epidemic data are characterized by strong randomness and show a typical non-stationary sequence, which poses a challenge to prediction. To mitigate the difficulty of prediction, we used VMD to reduce the complexity of the target sequence. Furthermore, we address the problem of empirically setting $K$ and the penalty factor in the VMD method by introducing the average sample entropy weighted by the correlation coefficient as the objective function. This enables the adaptive acquisition of $K$ and a penalty factor with PSO. In Section 3.1, we compare the VMD results with those of the CEEMDAN method, and find that the decomposition results of VMD method are superior to those of CEEMDAN.

In addition, as the development of the epidemic is influenced by multiple factors, the introduction of policy factors can better reflect the actual situation. The choice of the research interval determines the impact of the small-scale, precise city closure policy on the development of the epidemic. The sudden lifting of the ban one month before the unblocking caused a sudden influx of people, which could have led to fluctuations in the collection of epidemic data, which would thus not be representative. This data interval was therefore chosen.

Both the use of the VMD method and the introduction of multiple factors aimed to improve the accuracy of epidemic prediction. Initially, a single LSTM was used for prediction, but it did not meet the accuracy requirements. After combining with the VMD method, some improvement was observed, but there is still room for further enhancement. As a result, the PSO–VMD–AdaBoost–LSTM method was proposed in this paper, which achieved the best performance in terms of RMSE, MAE and MAPE compared to other models in the selected research area. Furthermore, the $R^2$ value was higher than that of other models, such as those presented in Tables 3 and 5.

While the proposed method provides a new approach to predicting the epidemic, some limitations exist due to the authors' knowledge and incomplete understanding of entropy theory. Further research is needed to address the following issues:

1.  The selection of models used to predict IMF components after VMD decomposition is limited, indicating the need to explore other models so as to optimize prediction accuracy;
2.  Due to the limited amount of epidemic data available, there is a possibility that the neural network's prediction accuracy could be affected;
3.  The growth of the epidemic is related to various factors, such as surrounding environment-related policies and population movement, but this paper only obtained factors related to containment policies. Therefore, the consideration of other factors is not comprehensive enough.

## 5. Conclusions

In this work, we apply VMD to the cumulative cases of COVID-19, alleviating the problem of mode mixing and high computational complexity encountered in CEEMDAN. To solve the defects of VMD itself, that is, the problem of the values of $K$ and $\alpha$, a method of optimizing VMD using PSO with inertia weight is proposed, that is, PSO–VMD. This method uses the correlation coefficient as the weight of the sample entropy to calculate the weighted average value, which is used as the fitness value to determine $K$ and $\alpha$. The adaptive values of $K$ and $\alpha$ are realized. In addition, to obtain a higher prediction accuracy, the AdaBoost method is used to enhance the LSTM. The results show that the AdaBoost boosting method can improve the performance of the LSTM network to a certain extent. Particularly when the prediction effect of the basic model is relatively general, the AdaBoost method is significantly improved. We used the cumulative confirmed data from Hubei Province and Hebei Province in China to verify and predict the epidemic situation over a period of 2 months. Finally, the model we proposed achieved the best results. The interval selected this time falls in the middle and late stages of the epidemic's development.

Currently, the epidemic's development is stable. In the forecast for Hubei Province, the fluctuation in the epidemic situation is very small, while the fluctuation projected in Hebei Province is slightly greater. Due to the adoption of stricter containment measures and precise targeted prevention and control measures in Hubei Province, the spread of the epidemic has been effectively curbed. In summary, this work has achieved good results in certain epidemic predictions, but there is still room for improvement. The VMD method plays a role in reducing the complexity of the data, and its application is not limited to epidemic prediction. It can also be extended to other fields, such as wind power and photovoltaics. Therefore, follow-up works should seek to obtain more different COVID-19 data, and train the model to strengthen its generalization ability. We should also try to use this model in different fields.

**Author Contributions:** Conceptualization, Zongyou Xia, Gonghao Duan and Ting Xu; methodology, Zongyou Xia and Gonghao Duan; software, Zongyou Xia; validation, Zongyou Xia; formal analysis, Gonghao Duan; data curation, Zongyou Xia; writing—original draft preparation, Zongyou Xia and Ting Xu; writing—review and editing, Zongyou Xia and Ting Xu. All authors have read and agreed to the published version of the manuscript.

**Funding:** This research is supported by the 111 project under Grant B17040 and Open Fund of Key Laboratory of Geological Hazards on Three Gorges Reservoir (China Three Gorges University), Ministry of Education under Grant 2022KDZ05. NO:CX2022363, Supported by Graduate Innovative Fund of Wuhan Institute of Technology.

**Data Availability Statement:** Publicly available datasets were analyzed in this study. This data can be found here: [https://github.com/OxCGRT/covid-policy-tracker], accessed on 13 April 2020.

**Conflicts of Interest:** The authors declare no conflict of interest.

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
