# Peer review of "Hybrid Prediction Model Based on Decomposed and Synthesized COVID-19 Cumulative Confirmed Data"

_ijgi, doi:10.3390/ijgi12060215_

Round 1

Reviewer 1 Report

Minors:
Fig.4 too small text (right side)
Table 3 R2 <= R^2
Fig.9 - Fig.10 please use identical colors between figures

Major:
The question is about stability of results (convergence to identical results).
Please repeat e.g. 10 times numerical experiment for different random seeds.

Reviewer 2 Report

The authors presented a deep learning approach to forecast the COVID-19 cases, However I have some concerns about the research:-

- The title named "trend prediction" is not scientific name, so please consider the types of cases that you predict.

- The paper didnot contain a literature review section.

- The results and comparison section is weak, so please consider the following references to deliver your comparison correctly 

- A Deep Learning BiLSTM Encoding-Decoding Model for COVID-19 Pandemic Spread Forecasting

- Forecasting the spread of the COVID-19 pandemic in Saudi Arabia using ARIMA prediction model under current public health interventions. 

- The contribution to use PSO optimizer is not clearly delivered.

- The conclusion cannot be stated as points.

Reviewer 3 Report

In this manuscript, the authors developed a hybrid model to predict COVID-19 trajectories with consideration of dynamic complexities. PSO-VMD was introduced given the limitations of empirical mode decomposition algorithms. Ensemble method was incorporated to improve the performance of individual LSTM networks. The model was then compared to several baselines to show its effectiveness. The approach is interesting but there are several weaknesses regarding model evaluation and method description. 

General comments

1. The authors selected the last 10% data for model validation. Though the model shows higher accuracy than other baselines, evaluation using that test set alone is insufficient to reflect prediction power. Data variations in the selected test set is extremely low (< 6 cases) and cannot be representative of the overall trend. Therefore, predictions for other time periods exhibiting more general trends must be evaluated.

2. Critical information on model development is missing. How can policy-based factors be integrated into the model? For ease of understanding, inputs, outputs and related hyperparameters of individual LSTM models should be specified. In addition, the significance of the multivariate dataset is not clearly explained. For example, can policy-based variables improve prediction performance? 

3. Compartmental modeling is a standard approach in epidemiology for robust prediction and quantification of impacts of different factors on COVID-19 spread (Pei et al., 2020; Chinazzi et al. 2020). Ren et al. (2021) discussed limitations of machine learning in COVID-19 forecasting (e.g., extrapolation). A potential issue is if machine learning can perform better than mechanistic modeling? The authors should specifically discuss advantages of data-driven hybrid modeling, given that epidemiological methods have performed well on disease modeling.

Specific comments

1. Line 149. More information on the PSO algorithm should be provided for clarity. For instance, what is the fitness function and how to set algorithm parameters?

2. Line 242. Please explain the advantage of VMD over CEEMDAN decomposition. It seems the former extracts fewer components.

3. Line 273. Are the predicted daily new cases obtained from a new model trained with daily new cases, or just calculated as the difference of predicted cumulative cases for neighboring days?

4. Please make sure the 9:1 data splitting strategy was used for validation; note that there are over 600 samples in total (Fig. 5) while the number of test samples is smaller than 50 (Fig. 8). 

5. Equation 1. The “min” operator is for the objective function instead of the constraint.

6. Equation 12. Start and end values of the summation are missing.

7. Is K in Eq. 1 the same as that defined in Eq. 14? Please clarify the differences between K and n defined in Eq. 17?

8. Moderate syntax and grammar errors need correction (e.g., Line 43, Line 160, Line 251, etc.).

References:

Chinazzi, M., Davis, J.T., Ajelli, M., Gioannini, C., Litvinova, M., Merler, S., y Piontti, A.P., Mu, K., Rossi, L., Sun, K. and Viboud, C., 2020. The effect of travel restrictions on the spread of the 2019 novel coronavirus (COVID-19) outbreak. Science, 368(6489), pp.395-400.

Pei, S., Kandula, S. and Shaman, J., 2020. Differential effects of intervention timing on COVID-19 spread in the United States. Science advances, 6(49), p.eabd6370.

Ren, X., Weisel, C.P. and Georgopoulos, P.G., 2021. Modeling Effects of Spatial Heterogeneities and Layered Exposure Interventions on the Spread of COVID-19 across New Jersey. International journal of environmental research and public health, 18(22), p.11950.

Round 2

Reviewer 1 Report

ok

Author Response

Please refer to the uploaded document

Reviewer 2 Report

The authors have enhanced the paper a lot. However, I have still some minor concerns:

1- Please add in the first lines in the abstract the covid-19 problem.

2- Section number one "Introduction" has a typo error.

3- Please seperate the introduction and the related works to two different sections.

4-I think that Equations 1,2,3,12,13 have miss typing like the others

5- The same issue in Figure 13, and 14 sizes have  a problem

Author Response

Please refer to the uploaded document

Reviewer 3 Report

The authors have addressed my concerns. I recommend acceptance of the manuscript after minor revision.

Eq. 1. “min" is for the objective function and the constraint after “s.t.” should be outside the curly bracket, which is the case in Dragomiretskiy and Zosso (2013).

There are several syntax and grammatical errors. Careful proofreading is highly recommended.

For example,

- Line 22. “method validated…” should be “method is validated…”

- Line 99. “the predicted the linear addition of the results to get the prediction results…” is difficult to understand

- Line 203. “In Figure 2 introduces…” should be “Figure 2 introduces…

- and so on.

Author Response

Please refer to the uploaded document
